# High APOBEC3B mRNA Expression Is Associated with Human Papillomavirus Type 18 Infection in Cervical Cancer

**DOI:** 10.3390/v14122653

**Published:** 2022-11-28

**Authors:** Gisele R. de Oliveira, Pedro S. Carvalho, Valdimara C. Vieira, Gislaine Curty, Diogo L. Basto, Miguel Ângelo M. Moreira, Marcelo A. Soares

**Affiliations:** 1Programa de Oncovirologia, Instituto Nacional de Câncer, Rio de Janeiro 20231-050, Brazil; 2Department of Immunology, Harvard Medical School, Boston, MA 02115, USA; 3Departamento de Genética, Universidade Federal do Rio de Janeiro, Rio de Janeiro 21941-901, Brazil; 4Programa de Genética, Instituto Nacional de Câncer, Rio de Janeiro 20231-050, Brazil

**Keywords:** APOBEC3B, cervical cancer, HPV, HPV types

## Abstract

The APOBEC3 (A3) proteins are cytidine deaminases that exhibit the ability to insert mutations in DNA and/or RNA sequences. APOBEC3B (A3B) has been evidenced as a DNA mutagen with consistent high expression in several cancer types. Data concerning the A3B influence on HPV infection and cervical cancer are limited and controversial. We investigated the role of A3B expression levels in cervical cancer in affected women positive for infection by different HPV types. Tumor biopsies from cancerous uterine cervix were collected from 216 women registered at Hospital do Câncer II of Instituto Nacional de Câncer, and infecting HPV was typed. A3B expression levels were quantified from RNA samples extracted from cervical biopsies using real-time quantitative PCR. Median A3B expression levels were higher among HPV18^+^ samples when compared to HPV16^+^ counterparts and were also increased compared to samples positive for other HPV types. In squamous cell carcinoma, HPV18^+^ samples also showed increased median A3B expression when compared to HPV Alpha-9 species or only to HPV16^+^ samples. Our findings suggest that A3B expression is differentially upregulated in cervical cancer samples infected with HPV18. A3B could be potentially used as a biomarker for HPV infection and as a prognostic tool for clinical outcomes in the context of cervical cancer.

## 1. Introduction

The human papillomaviruses (HPV) are composed of non-enveloped particles containing a double-stranded DNA genome of approximately 8 kb in size [1]. HPV is the causal agent of cervical cancer [2]. Over 200 HPV types have already been identified and classified into different genera according to their nucleotide sequence identity across the L1 open reading frame [3,4,5]. Among those genera, alphapapillomaviruses (αPV) are strongly associated with anal and cervical cancer development, particularly the α7 (HPV18-related) and α9 (HPV16-related) species [4,6]. αPVs also comprise the recognized high-risk (HR) HPVs based on their oncogenic potential, and HR HPV16 is the most prevalent among invasive cervical cancers worldwide, followed by HR HPV18 [7].

Cervical cancer is the fourth most incident cancer in women worldwide and in Brazil, responsible for over 8000 deaths in the latter in 2018 according to the last Globocan estimate [8]. Together with HPV, other host factors also influence cervical cancer risk and development. For example, increased expression of APOBEC3B was observed in high- and low-grade cervical lesions when compared to normal cytology, and its enhanced expression was directly associated with the mutational burden in cervical cancer tissues from affected subjects [9,10].

The APOBEC family is a group of enzymes with cytosine-to-uracil deaminase activity grouped into five classes, namely AID, APOBEC1, APOBEC2, APOBEC3 and APOBEC4. APOBEC3 is further subclassfied into seven members (A3A, A3B, A3C, A3D/E, A3F, A3G and A3H), characterized by a cytidine deaminase activity on single-stranded DNA [11,12]. Physiologically, these enzymes act as viral restriction factors and are also important for cellular RNA editing and somatic hypermutation, providing antibody diversification in B cells [12]. However, the overexpression of some APOBEC members has been associated with carcinogenesis in different body sites such as breast, uterus, bladder, lungs, head and neck [13,14,15]. In fact, members of APOBEC3 subfamily can act as DNA mutators through their cytidine deaminase activity [16]. Some studies showed that A3B expression is remarkably associated with a proliferative profile and suggested that A3B overexpression is an important mutational source for breast cancer development, highlighting its potential role in carcinogenesis [14,17].

A3B expression is consistently reported at low levels in normal human tissues [18]. However, it becomes overexpressed in neoplasia, being cervical cancer among the sites with the highest reported A3B levels [14]. Therefore, understanding the mechanisms underlying A3B overexpression during cancer onset is of paramount importance. In fact, a report from our group demonstrated that A3B mRNA levels were significantly enhanced in a keratinocyte cell line upon transfection with HR-HPV genomes [19], demonstrating a potential association between HPV and A3B overexpression. Hence, we aimed to investigate whether the presence of α7 and α9 HPV species differentially influences the expression of A3B in human cervical cancer samples.

## 2. Materials and Methods

### 2.1. Population

The population studied included 216 women with cervical cancer whose samples were extracted from uterine cervix biopsies. The subjects were registered and followed-up at Hospital do Câncer II of Instituto Nacional de Câncer José de Alencar Gomes da Silva (INCA). Tumor biopsies were collected at surgery rooms before patients initiated any therapy, and were stored in cryotubes with 1 mL of *RNA Later* (Life Technologies, California, EUA), conditioned in liquid nitrogen gallons and transported to Banco Nacional de Tumores e DNA (BNT) at INCA. After 24 h, the *RNA Later* was removed from the cryotubes and samples were placed in a −80 °C freezer until nucleic acid extraction. This study was approved by the Ethical Committee of INCA (registration # 156/10 and CAAE 53398416.0.0000.5274). An Informed Consent Term was applied to all the patients enrolled in the project.

All 216 subjects included in the study were previously investigated with respect to infecting HPV genotypes and HPV16/18 genetic variability [20,21]. All studied women were tested for concomitant HIV infection, and only two (0.9%) tested HIV-positive. Patients were selected for the current study based on the criteria of single infection with HPV16, 18, 31, 33, 35, 39, 45, 52, 56, 58, 59 or 73. Physicians reported patient clinical status during each appointment at their medical records. We consulted those records to collect demographic and clinical data, such as age at diagnosis, tumor subtype and tumor staging.

### 2.2. Total RNA Isolation and cDNA Synthesis

Total RNA extraction from the collected samples was performed using the All Prep DNA RNA kit (QIAGEN GmbH, Hilden, Germany), following the manufacturer’s protocol. After RNA isolation, samples were quantified using NanoDrop ND-1000 (Thermo Fisher Scientific, Waltham, MA, USA). RNA was kept on ice during all the procedures and stored at −80 °C. All RNA samples were treated with 1 µL RQ1 RNase-Free DNase (Promega, Madison, WI, USA), 1 µL RQ1 DNase 10X Reaction Buffer, 1–8 µL of input RNA (1–500 ng/µL) and ultrapure nuclease-free H_2_O, supplementing the final volume to 10 µL in order to degrade any DNA traces present in the samples. The mix was then incubated at 37 °C for 30 min and the DNase reaction was interrupted by adding 1 µL RQ1 DNase Stop Solution and proceeding to an incubation at 65 °C for 10 min. Continuing with the RNA treated with DNase, complementary DNA (cDNA) synthesis was performed as follows: 10 µL of input RNA after DNase treatment (1–500 ng/µL), 2 µL of 50 ng random primers (random hexameric primers—Thermo Fisher Scientific), 2 µL of 10 µM dNTPs (for each nucleotide) and ultrapure nuclease-free H_2_O complementing the final volume to 12 µL. The reaction was incubated at 65 °C for 5 min. After, 4 µL 5X First-Strand Buffer were added for each reaction, as well as 2 µL of 0.1 M DTT. The reaction was incubated at 25 °C for 1 min. For the last step, 2 µL of SuperScript™ II RT were added for each reaction following a cycling with 25 °C for 10 min, 42 °C for 50 min and 70 °C for 15 min.

### 2.3. Real-Time PCR for APOBEC3B mRNA

The relative quantification of *APOBEC3B* mRNA was performed through real-time PCR (qPCR). TaqMan™ probes for *GAPDH* (housekeeping gene, Hs02758991) and *APOBEC3B* (target gene, Hs00358981) were used, and the reactions were carried out in a ViiA 7 Real-Time PCR System apparatus (Thermo Fisher Scientific). Reactions were performed in triplicates containing 5 µL TaqMan™ Gene Expression Master Mix, 0.5 µL TaqMan™ probe for *APOBEC3B*, 0.25 µL TaqMan™ probe for *GAPDH*, 2 µL of input cDNA and ultrapure nuclease-free H_2_O, supplementing the final volume to 10 µL for each reaction. The cycling conditions were as follows: polymerase activation at 50 °C for 2 min, denaturation at 95 °C for 10 min and 40 cycles of denaturation at 95 °C for 15 s, followed by annealing and extension for fluorescence detection at 60 °C for 1 min. The analytical method used for calculating gene expression differences was the 2-ΔCT [22].

### 2.4. Statistical Analyses

The software *Statistical Package for the Social Sciences* (SPSS) was used for statistical analyses. The Mann–Whitney *U* test was used to analyze differences in *APOBEC3B* expression levels between the samples with distinct infecting HPV types and clinical variables (such as tumor histological type and clinical staging). Results were considered significant when *p*-values < 0.05. A multivariate analysis was also performed in the SPSS software using the Poisson regression model.

## 3. Results

In this study, we analyzed a total of 216 women with cervical cancer, all of them previously reported with single infection with different HPV types. The most frequent types observed were HPV16 (Alpha-9), present in 139 samples (64.4%) and HPV18 (Alpha-7), present in 30 samples (13.9%). Besides these, 10 additional HPV types were present in smaller frequencies. These included other members of the alpha-7 HPV species: HPV39 (*n* = 3; 1.4%), HPV45 (*n* = 17; 7.9%) and HPV59 (*n* = 4; 1.9%), and other members of the alpha-9 species: HPV31 and HPV35 (*n* = 3 each; 1.4%), and HPV33, HPV52 and HPV58 (*n* = 4 each; 1.9%). Moreover, we also found four samples positive for HPV73 (1.9%) and one positive for HPV56 (0.5%), belonging to the alpha-11 and alpha-6 species, respectively.

HPV lineages were identified in 131 (94.2%) of 139 HPV16^+^ and in 28 (93.3%) of 30 HPV18^+^ samples. Among the HPV16^+^ samples, ninety-two (66.2%) belonged to lineage A, twenty-eight (20.1%) to lineage D, six to lineage C (4.3%) and five to lineage B (3.6%). Among the HPV18^+^ patients, we also observed a higher frequency of lineage A in twenty-six samples (92.9%) and only two samples (7.1%) carried lineage B. HPV18 lineages C and D were not identified in this study. When we performed multiple intratype comparisons between samples carrying different lineages of HPV16 or HPV18 with respect to *A3B* expression levels, no significant differences were observed (*p* = 0.592 and *p* = 0.475, respectively).

Demographic and clinical characteristics of the patients and their relation to HPV are listed in Table 1. We found median ages of 45.5, 49, 49.5 and 50 years in women positive for HPV18, HPV16, Alpha-7 (HPV18 excluded) and Alpha-9 (HPV16 excluded), respectively. The overall median age was 49 years and did not differ significantly among the abovementioned groups (*p* = 0.753; Kruskal–Wallis). The most frequent histological type observed in our cohort was the squamous cell carcinoma (SCC) (84.8%). Women with HPV16 showed a significantly increased frequency of SCC (89.2%) when compared to those carrying HPV18 (56.7%) given the non-overlapping confidence intervals (respectively 84–94.4% and 38.2–75.2%). Conversely, HPV18-infected women exhibited a higher incidence of adenocarcinoma (ADC) (43.3%) in comparison to women with HPV16 (10.8%) without superposition between the confidence intervals (respectively 24.8–61.8% and 5.6–16%). Most patients presented at intermediary to late stages at diagnosis (mostly stages II and III, with 41.7% and 35.5%, respectively) and did not differ between HPV types in that regard.

In order to investigate the *APOBEC3B* expression levels in cervical cancer samples, we compared them among distinct HPV type groups. The median *APOBEC3B* expression level was higher among HPV18^+^ samples (*n* = 30) when compared to HPV16^+^ samples (*n* = 139) (*p* = 0.048; Figure 1). We also compared HPV18^+^ samples with all remainig HPV types identified in our study combined (except for HPV16). As expected, HPV18^+^ samples continued to exhibit significantly increased levels of *APOBEC3B* expression compared to that group (*p* = 0.027; Figure 1), suggesting that HPV18 is associated with a differential upregulation of *APOBEC3B* in human cervical cancer samples.

Next, we sought to analyze if *APOBEC3B* expression levels would remain significantly different at the HPV species phylogenetic level. A comparison of samples infected with HPV from Alpha-7 species with those infected with the Alpha-9 species was not significant (*p* = 0.377). On the other hand, HPV18^+^ samples continued to display an increase in the median *APOBEC3B* expression levels when compared to samples infected with the Alpha-9 (HPV16-related and included) species (*n* = 148) (*p* = 0.044, Figure 1). Moreover, as expected, the median A3B expression level was significantly lower in HPV16^+^ samples when compared to samples from Alpha 7 (HPV18-related and included) species (*n* = 54) (*p* < 0.001; Figure 1).

Differences in *APOBEC3B* expression levels were also observed considering only SCC samples. Again, HPV18^+^ samples (*n* = 30) showed an increased median *APOBEC3B* expression level when compared to Alpha-9 species (*n* = 148) (*p* = 0.021; Figure 1). More specifically, SCC HPV-18^+^ samples exhibited an increase in the median A3B expression degree when compared to SCC HPV16^+^ samples (*p* = 0.024; Figure 1).

With respect to the clinical variables under study, we did not find any significant association between A3B expression and tumor subtypes or clinical staging using univariate analyses. However, when analyzing age at diagnosis, we found that women above the median age (49 years) displayed higher A3B expression levels than those under the median (*p* = 0.013). That prompted us to analyze the effect of each variable in the A3B expression levels. A final model was tested in a multivariate analysis in order to simultaneously consider all variables (HPV type, histological subtype and clinical staging). After multivariate testing (Table 2), only HPV type was significantly associated with A3B expression levels (e.g., HPV18^+^ with higher A3B expression; *p* = 0.011 and HPV31^+^ and HPV56^+^ with lower A3B expression; *p* < 0.001).

## 4. Discussion

In this study, we describe that *APOBEC3B* expression is differentially upregulated in human cervical cancer depending on the infecting HPV type. In fact, *APOBEC3B* exhibited an upregulation pattern in breast and ovarian cancer cell lines, demonstrating a consistent association between this cytidine deaminase activity and neoplasia [23,24]. Moreover, analysis of multiple malignancies with data deposited in *The Cancer Genome Atlas* (TCGA) showed the uterine cervix to be one of the cancer sites with the greatest A3B expression and APOBEC-related mutational signatures [9,14]. In this scenario, given the necessary role of HPV infection in cervical cancer development, an association between A3B overexpression and viral infection was considered. Here, we reported that HPV18^+^ cervical cancer samples exhibited an increased A3B mRNA expression when compared to HPV16^+^ samples (*p* = 0.027). These results not only reinforce the association between HPV and A3B expression in cervical cancer, but also suggest that distinct HPV types can modulate A3B expression to different extents.

Our efforts were concentrated in A3B, and not in other members of the APOBEC3 subfamily, for a number of reasons. In addition to the abovementioned evidence of the importance of A3B in multiple cancers (including cervical cancer), Burns et al. [14] showed that cervical cancers display high levels of cytosine mutational loads in the A3B-preferred trinucleotide context, suggesting a predominant role of A3B regarding this malignancy. Moreover, we have previously analyzed the expression of different APOBECs in vitro, including 3A and 3H, from immortalized keratinocytes infected with HPV16 and HPV18 [19], and only A3B showed significant expression upon infection by both HPV types.

High-risk persistent HPV infection is well established as the main risk factor for the development of premalignant intraepithelial lesions and cervical cancer progression [25,26,27]. Albertini and colleagues have demonstrated, in a persistent HPV18 infection model that the virus-induced immunosuppression was associated with the upregulation of nuclear factor kappa B (NF-kB) [28], which was described as an A3B expression inductor [24].

The data presented herein are consistent with those from previous studies that showed an increase in A3B mRNA levels upon HPV infection [29,30]. Vieira and colleagues demonstrated, in an immortalized keratinocyte cell line, that the transfection with high-risk HPV16 or HPV18 genomes upregulated A3B mRNA levels, and that the transfection with E6 gene from those types alone was sufficient for the observed A3B upregulation [19]. HPV18 transfection conferred higher levels of A3B induction, with a five- to ten-fold increase compared to negative controls. Its transfection also stimulated greater deaminase activity of about five-fold relative to negative controls [19]. Moreover, human foreskin keratinocytes carrying HPV-16 or HPV-18 episomal genomes displayed increased levels of A3B mRNA expression, whereas higher levels were noticed among HPV18^+^ samples [31]. In normal breast epithelial cells transfected with HPV18, induced overexpression of A3B was observed, but when HPV18 E6/E7 were silenced, A3B induction was abrogated. Therefore, A3B is apparently activated through HPV-18 E6/E7 [32].

A recent report evaluated the potential mechanistic link between metastatin-associated protein 1 (MTA1), whose levels are tightly associated with tumor progression in cervical cancer, and A3B in cervical cancer cells. It was observed that HPV18 infection was significantly associated with *MTA1* mRNA levels and HPV18 E6 could induce its expression. It was also evidenced that transfection of the HPV18 genome induced an increase in DNA deaminase activity, which was proportional to A3B upregulation and required MTA1 participation [33].

Although a mechanistic model linking HPV to A3B upregulation is notfully established, recent reports suggest promising pathways. Higher A3B mRNA levels were observed among *TP53* mutated breast cancer samples [23]. The p53 is a known target of HPV E6-mediated ubiquitination and subsequent degradation [5]. Conversely, p53 induces p21 and recruits the DREAM repressive complex at the APOBEC3B promoter site, reducing its expression. Therefore, *TP53* loss or E6-mediated degradation of p53 prevents the inhibition pathway and possibly leads to A3B upregulation [34]. Moreover, the HPV E7 oncoprotein can also be involved in viral-mediated A3B induction. E7 has been suggested to disrupt the p53-p21-DREAM signaling, preventing the inhibition of several DREAM-targeted genes such as *APOBEC3B* [35]. In this scenario, it is plausible to speculate that both E6 and E7 oncoproteins effectively abrogate an important repressive control of A3B transcription, which could be a mechanism exploited during HPV infection that leads to A3B upregulation. Nevertheless, other pathways besides the DREAM signaling seem to be associated with HPV-induced A3B overexpression. In an immortalized keratinocyte cell line, HPV16 E6 was able to upregulate A3B upon the induction of TEAD transcription factors that directly bind to its promoter. However, while TEAD knockdown prevented HPV16-mediated A3B induction in keratinocyte cell lines, as expected, this was not observed for cervical cancer cell lines (CaSki and HeLa), suggesting that, at least for cervical cancer, other pathways are involved in A3B upregulation [31].

Another possible link between HPV and A3B overexpression could be through the antiviral responses elicited by the innate immunity. During HPV infection of human keratinocyte cell lines, the viral E5 protein induces the expression of IFN-β [36]. Either type I (including IFN-β) or type II interferon molecules induce antiviral responses and control viral replication. To bind to its receptor, IFNs trigger signaling cascades that lead to the induction of several targeted genes [37]. In fact, APOBEC3F and APOBEC3G were identified among the interferon stimulated genes with a direct antiviral activity [38]. Therefore, it is possible that HPV-mediated interferon secretion could stimulate APOBEC3 transcription. However, the specific APOBEC3 enzyme stimulated by the IFN cascade seems to vary according to the analyzed tissue. In human blood monocytes, IFN-α was shown to stimulate APOBEC3A, APOBEC3F and APOBEC3G expression, with the last being especially relevant to the anti-HIV-1 activity, but had no consistent effect on APOBEC3B and APOBEC3C mRNA levels [39]. Upon IFN-β stimulation, W12 cells (cervical keratinocytes that retain an HPV16 episomal form) upregulated A3A, A3F and A3G, but not A3B [40]. Those results suggest that, although an association between viral infection, IFN secretion and APOBEC3 upregulation is plausible, it is not likely the case for A3B in cervical cell lines and in primary monocytes. However, IFN-α was able to upregulate A3B mRNA levels as well as A3C, A3F and A3G in primary human hepatocytes, indicating that IFN-mediated APOBEC3 upregulation is highly variable across different tissues and does not seem applicable to our findings in human cervical cancer samples [41].

Taken together, different reports indicate A3B upregulation upon HPV infection. However, APOBEC3 deaminase activity can also target the HPV genome, induce hypermutation and limit its infectivity, contributing to virus clearance [10,40,42]. In fact, a concordance between host and viral APOBEC-related mutations has been established for HPV^+^ oropharyngeal squamous cell carcinoma [43]. Therefore, a question that remains unanswered is whether the HPV-dependent A3B induction contributes to HPV carcinogenesis or limiting viral infectivity. In fact, although both A3A and A3B were upregulated by HPV16 E7 in keratinocytes, only A3A was able to abrogate HPV infection, which could suggest that the higher levels of A3B do not significantly impact HPV genomes [29]. Interestingly, APOBEC3-mediated viral restriction exerted selective pressure on papillomavirus genomes throughout its evolution leading to a significant depletion of TC dinucleotides (an important target sequence for human APOBEC3 proteins). The most remarkable TC dinucleotide depletion was found among alphapapillomaviruses infecting the anogenital mucosa, a tissue with high levels of expressed APOBEC3 proteins, suggesting that APOBEC3 contributed to papillomavirus evolution selecting variants with reduced TC content and, therefore, less sensitive to its viral restriction capacity [10].

Our study is limited in the sense that it could not unveil the molecular determinants that govern the differential effect of HPV18 towards A3B upregulation when compared to HPV16 or other high-risk HPV types. We have previously shown that HPV18 infection of human immortalized keratinocytes caused higher levels of A3B induction (1.5-fold) in comparison to HPV16 [19]. In the same study, we analyzed whether the expression of E6 was sufficient to induce A3B upregulation. Interestingly, only cells expressing high-risk E6 proteins showed significant increases in A3B mRNA levels in comparison to control and E6 from low-risk HPV types, and HPV18 E6 showed higher increases in A3B expression compared to the HPV16 counterpart. A recent report has evidenced that HPV18-associated premalignat lesions and invasive cancers have increased mutated viral E4 genes as resulted from APOBEC-related activity compared to other viral genes and to HPV16 counterparts [44]. Althogether, our results add to evidence suggesting that distinct viral and host genomic alterations occur between HPV16 and HPV18 infections, possibly reflecting differences in cell transformation mechanisms induced by these two HPV genotypes, but further experimentation is required to elucidate the mechanisms behind those differences.

In conclusion, we found that high APOBEC3B expression was associated with HPV18 infection in women with cervical cancer. Our results suggest that A3B expression is an important biomarker of HPV infection, and potentially a prognostic biomarker of cervical cancer clinical outcomes such as recurrence, response to treatment and overall survival, issues that will require further investigation.

## Figures and Tables

**Figure 1 viruses-14-02653-f001:**
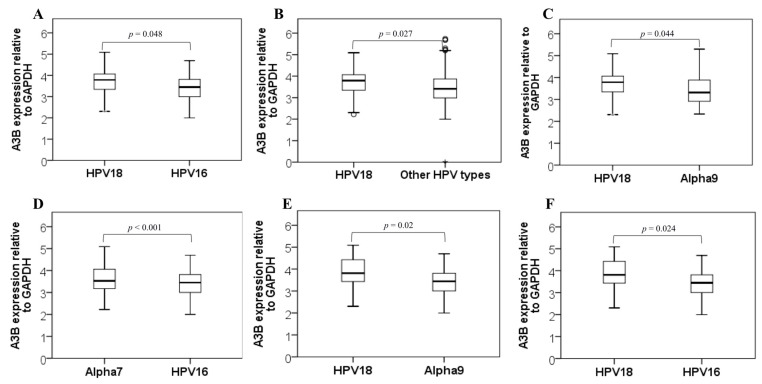
APOBEC3B mRNA expression relative to GAPDH in cervical cancer samples comparing (**A**) HPV18 and 16; (**B**) HPV18 and all other HPV types; (**C**) HPV18 and Alpha-9 species; and (**D**) HPV16 and Alpha-7 species (**D**; *p* < 0.001). APOBEC3B mRNA expression relative to GAPDH in squamous cell carcinoma (SCC) samples comparing HPV18^+^ with Alpha-9 species (**E**) and with HPV16-infected women (**F**).

**Table 1 viruses-14-02653-t001:** Clinical characteristics of women with cervical cancer followed in the study according to infecting HPV type (INCA-RJ).

		HPV16 (*n* = 139)		HPV18 (*n* = 30)		Alpha-7 * (HPV18 Excluded) (*n* = 24)		Alpha-9 ** (HPV16 Excluded) (*n* = 18)		Total	
Tumor Histological Type	ADC	10.8	5.6–16	43.3	24.8–61.8	12.5	0–26.5	5.6	0–16.9	15.2	10.3–20
	SCC	89.2	84–94.4	56.7	38.2–75.2	87.5	73.5–100	94.4	83.1–100	84.8	80–89.7
Staging	I	18.7	12.2–25.2	16.7	2.8–30.6	12.5	0–26.5	16.7	0–35.2	17.5	12.4–22.7
	II	41.7	33.5–50	36.7	18.7–54.7	50	28.9–71.1	38.9	14.6–63.1	41.7	35–48.4
	III	33.1	25.2–41	43.3	24.8–61. 8	33.3	13.4–53.2	44.4	19.7–69.2	35.5	29–42
	IV	6.5	2.3–10.6	3.3	0–100	4.2	0–100	0	-	5.2	0–11.7

SCC, squamous cell carcinoma; ADC, adenocarcinoma. * Members of the HPV Alpha-7 species included: HPV39, HPV45 and HPV59. ** Members of the HPV Alpha-9 species included: HPV31, HPV33, HPV35, HPV52 and HPV58.

**Table 2 viruses-14-02653-t002:** Multivariate analysis for the association of APOBEC3B expression levels with HPV type, histological type and clinical staging at diagnosis in women with cervical cancer from INCA-RJ.

Variables	Categories	Adjusted *p*-Value	OR	IC95%
**HPV Type**	HPV16		1	
	**HPV18**	**0.011 ***	**1.26**	**1.05–1.50**
	**HPV31**	**0.001**	**0.46**	**0.29–0.73**
	HPV33	0.096	1.28	0.95–1.71
	HPV35	0.233	1.27	0.85–1.90
	HPV39	0.408	1.21	0.76–1.90
	HPV45	0.893	0.98	0.77–1.25
	HPV52	0.821	0.93	0.50–1.71
	**HPV56**	**<0.001**	**0.65**	**0.52–0.81**
	HPV58	0.714	0.87	0.41–1.83
	HPV59	0.664	0.89	0.52–1.50
	HPV73	0.088	0,91	0.82–1.01
**Tumor Type**	SCC		1	
	ADC	0.564	0.94	0.76–1.15
**Stage**	I	0.344	0.84	0.57–1.15
	II	0.652	0.92	0.67–1.28
	III	0.412	0.87	0.63–1.20
	IV		1	

SCC, squamous cell carcinoma; ADC, adenocarcinoma. * Significant *p*-values at the 0.05 level are in boldface.

## Data Availability

Not applicable.

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
