# Peer review of "High APOBEC3B mRNA Expression Is Associated with Human Papillomavirus Type 18 Infection in Cervical Cancer"

_viruses, 2022, doi:10.3390/v14122653_

Round 1

Reviewer 1 Report

The manuscript titled “High APOBEC3B mRNA expression is associated with human 2 papillomavirus type 18 infection in cervical cancer” by Oliveira et al. analyzes APOBEC3B mRNA expression of 216 women with cervical cancer and indicates an association between high A3B expression and HPV type 18. The results presented in the manuscript are clear and informative. Below are suggestions that can be addressed with further improve the manuscript.

1. The differences in A3B expression between HPV18 and HPV16/Apha9/other, although significant, are not very drastic. To provide the readers a better context, how do A3B expression levels of these HPV-positive tissue compare with those of the matched normal tissue? Considering the heterogeneity of these tissue samples, have the authors performed histology to evaluate the A3B expression in addition to qPCR?

2. Have the authors determined expression levels of APOBEC3A and APOBEC3H in these samples, as they have also been implicated in cancer mutagenesis?

3. Since the main results of the manuscript is HPV18, compared with other types, is more associated with high A3B expression, it would be great if the authors could provide possible explanations as well as clinical significance of this association. The current version of the discussion focuses heavily on the link between HPV and A3B but falls short on specifically HPV-18 and A3B. What possibly explains why HPV-18 is more strongly associated with A3B than HPV-16?

Author Response

The manuscript titled “High APOBEC3B mRNA expression is associated with human papillomavirus type 18 infection in cervical cancer” by Oliveira et al. analyzes APOBEC3B mRNA expression of 216 women with cervical cancer and indicates an association between high A3B expression and HPV type 18. The results presented in the manuscript are clear and informative. Below are suggestions that can be addressed with further improve the manuscript.

  1. The differences in A3B expression between HPV18 and HPV16/Apha9/other, although significant, are not very drastic. To provide the readers a better context, how do A3B expression levels of these HPV-positive tissue compare with those of the matched normal tissue? Considering the heterogeneity of these tissue samples, have the authors performed histology to evaluate the A3B expression in addition to qPCR?

 Several previous studies have suggested low levels or absence of A3B expression in most normal tissues (Refsland et al. 2010, 10.1093/nar/gkq174; Burns et al. 2013, 10.1038/nature11881; Bruns et al. 2013, 10.1038/ng.2701). As discussed in the current manuscript (pg 7, line 3), E6 and E7 HPV proteins are able to upregulate A3B expression, and when their expression was silenced, A3B expression was abrogated, yet the detailed mechanisms of this regulation are not well defined.

With respect to the use of other techniques for quantifying A3B expression, we used real-time PCR, which is the gold standard for gene expression quantification. The development of A3B antibodies, especially for immunohistochemistry usage, has been challenging. Until recently, no A3B-specific antibodies had been developed, and the group by Dr. Reuben Harris has developed one monoclonal antibody in 2019 (Brown et al 2019; 10.3390/antib8030047). This mAb is supposedly suitable for immunohistochemistry, but we did not have a chance to test it in our tissue samples.

  1. Have the authors determined expression levels of APOBEC3A and APOBEC3H in these samples, as they have also been implicated in cancer mutagenesis?

Despite the fact that APOBEC3A and 3H have also been implicated in cancer mutagenesis, we focused on APOBEC3B, whose expression is associated with increased risk of breast cancer and is also described as an important risk factor for cervical cancer development (Burns et al 2013, 10.1038/ng.2701; Cescon et al. 2015; 10.1073/pnas.1424869112). Burns et al 2013 (10.1038/ng.2701) showed previously that cervical cancers are among the tumor types that display the highest levels of A3B expression and that they also display high levels of cytosine mutational loads in the A3B-preferred trinucleotide context, suggesting a predominant role of A3B on cervical cancer. Moreover, we have previously analyzed the expression of different APOBECs in vitro, including 3A and 3H, from immortalized keratinocytes infected with HP16 and HPV18 (10.1128/mBio.02234-14), and only A3B showed significant expression upon infection by both HPV types. We have added such issues to the Discussion of the revised manuscript.

  1. Since the main results of the manuscript is HPV18, compared with other types, is more associated with high A3B expression, it would be great if the authors could provide possible explanations as well as clinical significance of this association. The current version of the discussion focuses heavily on the link between HPV and A3B but falls short on specifically HPV-18 and A3B. What possibly explains why HPV-18 is more strongly associated with A3B than HPV-16?

We have shown previously that HPV18 infection of human immortalized keratinocytes caused higher levels of A3B induction (1.5-fold) in comparison to HPV16 (10.1128/mBio.02234-14). In the same study we analyzed whether expression of E6 was sufficient to induce A3B upregulation. Interestingly, only cells expressing high-risk E6 proteins showed significant increases in A3B mRNA levels in comparison to control and E6 from low-risk HPV types, and again HPV18 E6 showed higher increases in A3B expression compared to the HPV16 counterpart. Recent evidence further points out to differences in the genetic effects of A3B mutation loads in HPV18 genomes compared to HPV16 (10.1016/j.tvr.2021.200221; already cited in the original manuscript). Therefore, we hypothesize that genetic differences between distinct HPV types may be responsible for the results seen in the present report, but further experimentation is required to elucidate the mechanisms behind those differences. We have added such piece of information to the Discussion of the revised manuscript.

Reviewer 2 Report

In the article by de Oliveira, et. al., an epidemiological study was conducted in which the prevalence of HPV types was evaluated in 216 cervical cancer samples, with HPV 16 being the most prevalent type. In addition, the prevalence of HPV lineages was analyzed. Based on clinical features, HPV-16 was mainly associated with squamous cell carcinoma (SCC), while HPV-18 was associated with adenocarcinoma (ADC). Interestingly, the study shows that APOBEC3B (A3B) expression is significantly overexpressed in HPV18-containing tumors, compared to HPV16-positive tumors and other tumors with different HPV types. Furthermore, A3B expression was found not to be associated with the different HPV 16 and 18 lineages. In agreement, when only SCC samples were analyzed, an increase in A3B was observed in those tumors positive for HPV18. No association of A3B expression with tumor subtype or clinical stage was found. Although the article is interesting and provides novel information on the association of APOBEC3B expression with HPV18-positive tumors, it is restricted to an epidemiological study. It would be interesting to evaluate these findings in HPV-negative, HPV18-positive cell lines derived from cervical cancer. other viral types; and to evaluate, through in vitro studies, the deregulation of A3B by the viral oncoproteins of HPV18 and its participation in processes associated with cancer. In addition, the study only focuses on proposing A3B as a biomarker of infection, which is limited since it would be valuable to evaluate the potential use of A3B as a prognostic biomarker through A3B expression and its association with the clinical outcome of cervical cancer patients such as recurrence, response to treatment and overall survival.

Author Response

In the article by de Oliveira, et al., an epidemiological study was conducted in which the prevalence of HPV types was evaluated in 216 cervical cancer samples, with HPV 16 being the most prevalent type. In addition, the prevalence of HPV lineages was analyzed. Based on clinical features, HPV-16 was mainly associated with squamous cell carcinoma (SCC), while HPV-18 was associated with adenocarcinoma (ADC). Interestingly, the study shows that APOBEC3B (A3B) expression is significantly overexpressed in HPV18-containing tumors, compared to HPV16-positive tumors and other tumors with different HPV types. Furthermore, A3B expression was found not to be associated with the different HPV 16 and 18 lineages. In agreement, when only SCC samples were analyzed, an increase in A3B was observed in those tumors positive for HPV18. No association of A3B expression with tumor subtype or clinical stage was found. Although the article is interesting and provides novel information on the association of APOBEC3B expression with HPV18-positive tumors, it is restricted to an epidemiological study. It would be interesting to evaluate these findings in HPV-negative, HPV18-positive cell lines derived from cervical cancer. other viral types; and to evaluate, through in vitro studies, the deregulation of A3B by the viral oncoproteins of HPV18 and its participation in processes associated with cancer. In addition, the study only focuses on proposing A3B as a biomarker of infection, which is limited since it would be valuable to evaluate the potential use of A3B as a prognostic biomarker through A3B expression and its association with the clinical outcome of cervical cancer patients such as recurrence, response to treatment and overall survival.

As already described in the response to comment #3 by Reviewer #1, we have previously shown (Vieira et al 2014; 10.1128/mBio.02234-14) that normal immortalized (HPV-negative) cell lines only express negligible levels of A3B, but that E6/E7 from HPV18 is required to cause A3B upregulation. Other HPV types of the alpha-7 and alpha-9 species were also tested in vitro in that study and also from natural infections in the present study.

With respect to the potential use of A3B expression as a prognostic biomarker to clinical outcomes and response to treatment, we agree with the Reviewer, and added that to the revised Discussion section. Additional studies, however, are required to elucidate such potential uses.

Round 2

Reviewer 2 Report

In the current version, the authors have revised the manuscript based on previous comments.

Moreover, typos and spelling errors need correction. For example:

on line 235, change the word "APOBEEC3" to "APOBEC3".

on line 240, change the word "HP16 to HPV16".